# Assessment of Fire Regimes and Post-Fire Evolution of Burned Areas with the Dynamic Time Warping Method on Time Series of Satellite Images—Setting the Methodological Framework in the Peloponnese, Greece

**Nikos Koutsias** *, **Anastasia Karamitsou, Foula Nioti and Frank Coutelieris**

Department of Environmental Engineering, University of Patras, GR-30100 Agrinio, Greece
* Correspondence: nkoutsia@upatras.gr; Tel.: +30-26410-74201

**Abstract:** Forest fires are considered to be an important part of numerous terrestrial ecosystems and vegetation types, being also a significant factor of ecosystem disruption. In this sense, fires play an important role in the structure and function of the ecosystems. Biomes are characterized by a specific type of fire regime, which is a synergy of the climate conditions and the characteristics of the vegetation types dominating each biome. The assessment of burned areas and the identification of the fire regimes can be implemented with freely available low- to high-resolution satellite data as those of Landsat and Sentinel-2. Moreover, the biomes are characterized by the phenology, a useful component for vegetation monitoring, especially when time series of satellite images are used. Both the identification of fire regime by reconstructing the fire history and the monitoring of the post-fire evolution of burned areas were studied with remote sensing methods. Specifically, the present paper is a pilot study implemented in a Mediterranean biome, aimed at establishing the methodological framework to (i) define fire regimes, (ii) characterize the phenological pattern of the vegetation (pre-fire situation) of the fire-affected areas, and (iii) compare the phenology of the recovered fire-affected areas with the corresponding one of the pre-fire situation. At the global level, based on MODIS fire perimeters, we found that fires are occurring at 70% in the tropical and subtropical grasslands, savannas, and shrublands, followed by fires at tropical and subtropical moist broadleaf forests by 7% and by fires at deserts and xeric shrublands by 6.5%.

**Keywords:** fire regimes; post-fire recovery; burned areas; vegetation phenology; time series; satellite images; satellite remote sensing; dynamic time warping; Mediterranean





## 1. Introduction

Fire has always been a way for human beings to manage both natural and artificial ecosystems [1,2]. There is a mutual relationship between fire activity and ecosystems at various scales, through the regulation of ecosystem dynamics and the carbon cycle [1], supporting the hypothesis of the interaction between patterns and processes. Forest fires are an important part of numerous terrestrial ecosystems and vegetation types [3,4], but they are also a significant factor of ecosystem disruption. Thus, fires play an important role in the ecosystems' functionality and in the identification of their biodiversity patterns [5]. In the existing relative literature, it is often assumed that fires of various sizes and locations favor the creation of a vegetation mosaic [6]. However, the role of fire in landscape heterogeneity is rather complex since spatial heterogeneity depends on scale [7]. Fire generally has a dual role in the pattern of the landscape, either as a mechanism of homogenization on a small scale or as a mechanism of differentiation on a larger scale [8]. However, the relationship linking fire history, vegetation succession, and landscape composition is poorly understood in some vegetation types [9].

The biomes and the associated climate zones are characterized by a specific type of fire regimes, which can be identified by the historical fire records of an area. Beyond the human component, the fire regime is a synergy of the climate conditions and the characteristics of the vegetation types that are dominant in each biome [10–12]. Moreover, the biomes are characterized by the phenology of the vegetation types, representing a useful characteristic for the monitoring of the vegetation, especially when remote sensing methods are involved [13,14]. Both the identification of the fire regime by reconstructing the fire history of an area [15,16] and the monitoring of the post-fire evolution of burned areas [17] can be studied with existing remote sensing methods using time series of satellite images as main data source [18].

The relationship between the expansion of vegetation and the climate on a global scale can be described by analyzing the expansion of biomes whose concept is an approach to organizing the ecological diversity on a large scale. The terrestrial plant formations are classified on the basis of the dominant plant functional groups [19], and the number of identified plant formations varies depending on the classification system adopted. The distribution of vegetation, although not completely identical, is related to the various climatic classifications proposed, such as the Köppen–Geiger classification (adjusted by Kottek et al. [20]). This confirms the strong relationship between the spatial distribution of macroclimatic parameters and the spread of vegetation on a global scale [21]. Moreover, biogeographical models correspond to the expansion of potential vegetation with climatic variables that describe the energy and water balance—mainly temperature and rainfall [22], as well as soil parameters [19].

It is, thus, understood that the climatic parameters which determine the fire regime also determine the special characteristics of the various plant formations and, accordingly, the plant formations are characterized by different fire regimes [23]. This differentiation can be distinguished at various spatial scales, from forest formations at national [24] and continental level [25] to vegetation on a global scale [26]. For instance, in the coniferous forests in northern areas, the period of recurrence of natural fires (fire rotation) is estimated at 50–500 years [10,27,28], while the same variable in tundra can be estimated to up to 7000 years [27]. Conversely, in savannas, where the rapid and intense growth of fine biomass is favored [29], extensive low-intensity fires can occur up to twice a year [11]. The Mediterranean ecosystems occupy an intermediate position showing more fires of a smaller extent in comparison to those of the northern forests, but less than those of savannah. So far, relatively few studies [30–33] have investigated fire regimes comparing different vegetation and climatic zones, as well as their time changes, although it is known that the synthesis of fire statistics from many different types of fires together with climate may obscure the relationship linking climate, fuel type, and fire regime [34].

The methodology developed and presented here aims to define fire regimes, as well as to assess post-fire patterns of burned areas in selected biomes across the planet by studying the landscape phenology with time series of satellite images. The availability at no cost to the final user of (i) Landsat satellite imagery by the US Geological Survey (USGS), (ii) Sentinel-2 satellite imagery, provided by ESA, and (iii) MODIS (or Moderate Resolution Imaging Spectroradiometer) imagery provided by NASA/USGS, allows for a low-cost obtaining and processing of time series of satellite images (i.e., 1984–today). Under other circumstances, this would be a considerable cost that could be prohibitive to research. In the current work, the methodology is applied in the Mediterranean biome as represented by the Peloponnese, Greece.

Specifically, the research objectives applied consist of three distinct parts:

1. Spatially explicit reconstruction of the recent fire occurrence history starting from 1984 and identification of the fire regimes with Landsat and Sentinel-2 satellite data.
2. Identification and description of the phenology of the pre-fire vegetation using spectral bands and vegetation indices from time series of MODIS satellite images.

3.      Observation and comparison of post-fire evolution patterns of burned areas by comparing the phenology of the fire-affected areas with the phenology of the vegetation before the fire using time series of MODIS satellite images.

## 2. Material and Methods

The implementation of this pilot research study includes five distinct parts. The first two, belonging to the materials section, are (i) the sampling design that concerns the organization and selection of sampling areas, and (ii) the acquisition of time series of satellite images including the selection, download, and pre-processing of the required satellite images. The last three, belonging to methods section, are (iii) the reconstruction of recent fire history that is the mapping of burned areas and the determination of fire regimes with Landsat and Sentinel-2 satellite data, (iv) the determination of vegetation phenology in the pre-fire situation using satellite observations, and (v) the monitoring of post-fire recovery patterns in the selected biomes by assessing the vegetation phenology of fire affected areas with MODIS satellite data.

### 2.1. Materials

2.1.1. Sampling Design

We adopted the classification system proposed by Olson et al. [35], which identifies and describes 14 plant formations (Figure 1). The climatic conditions that characterize each plant form also affect the composition and structure of vegetation while they determine the fire regimes. Out of all the identified formations, fires are common in some of them, while, in others, they are very rare, either due to the lack of fuel or because of climatic constraints. By using archived data of global fire incidents resulting from the analysis of MODIS satellite data [36] and by implementing spatial analysis methods including the kernel density interpolation method [37], global fire density maps have been created worldwide to identify fire-affected areas. The kernel density interpolation method was first used in forest fires by Koutsias, Kalabokidis, and Allgöwer [37], in order to minimize the uncertainty of the location of forest fire outbreaks, and has been applied to various topics including the creation of forest fire zones at multiple scales [38–40].

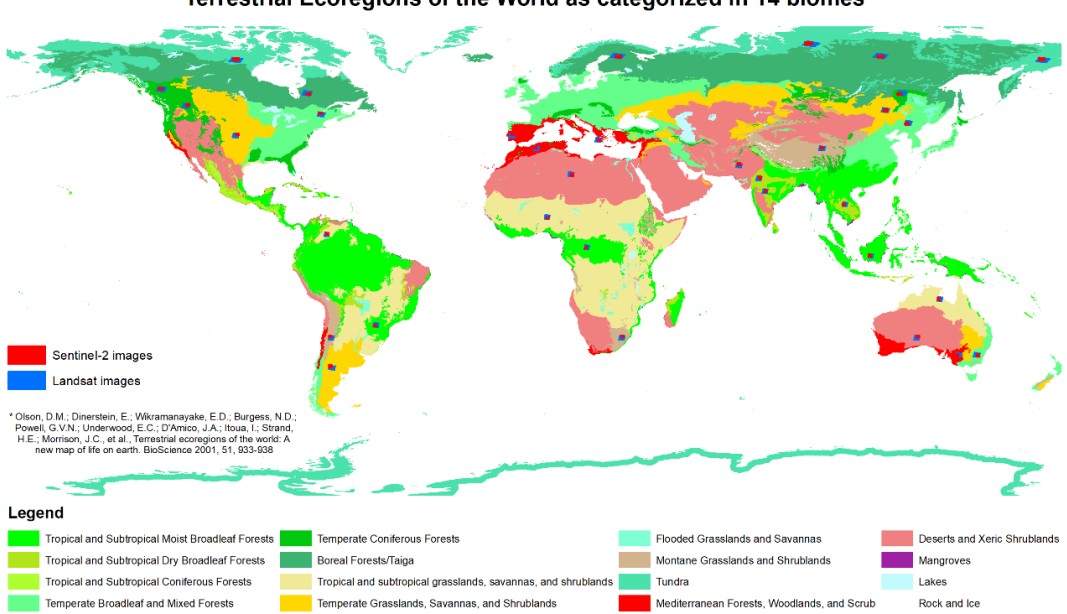

**Figure 1.** The ecoregions are categorized within 14 biomes as proposed by Olson et al. [35], together with the Landsat and Sentinel-2 scenes chosen to reconstruct the recent fire history with satellite data. Data were downloaded from https://www.worldwide.org/publications/terrestrial-ecoregions-of-the-world (accessed on 1 September 2017), and the original symbols were used to make the map.

By taking into account the fire-affected areas, 1–3 frames (path/row) were selected from the Worldwide Reference System (WRS-2) corresponding to a LANDSAT scene and a Sentinel-2 granule that coincides to each Landsat scene for each biome. Each of these frames corresponds to a satellite image of Landsat with dimensions of 175 × 180 km or to a Sentinel-2 granule with dimensions of a 100 × 100 km, in which the recent fire history with Landsat and Sentinel-2 images was reconstructed. The selection of multiple sampling areas per biome (1–3) ensures that the reconstruction of the fire history will take into account some of the geographical differentiation of vegetation (i.e., Mediterranean vegetation spreads to Europe, North and South America, Africa, and Australia). For the Landsat images, a square of 100 × 100 km centered in each Landsat scene was considered to reconstruct the recent fire history as shown in Figure 2.

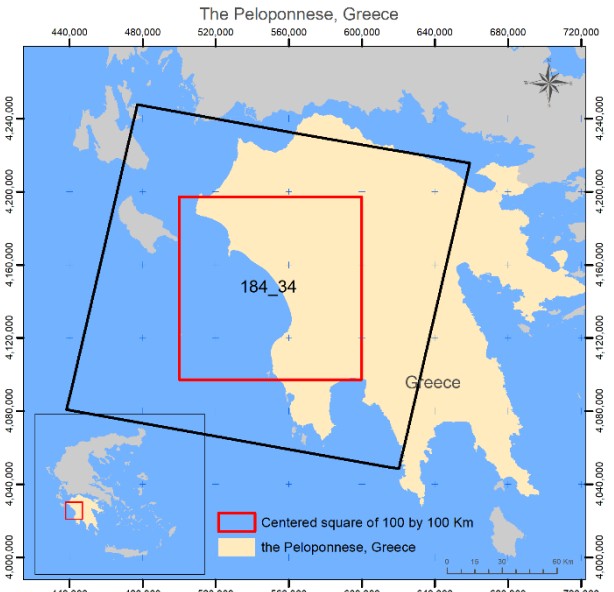 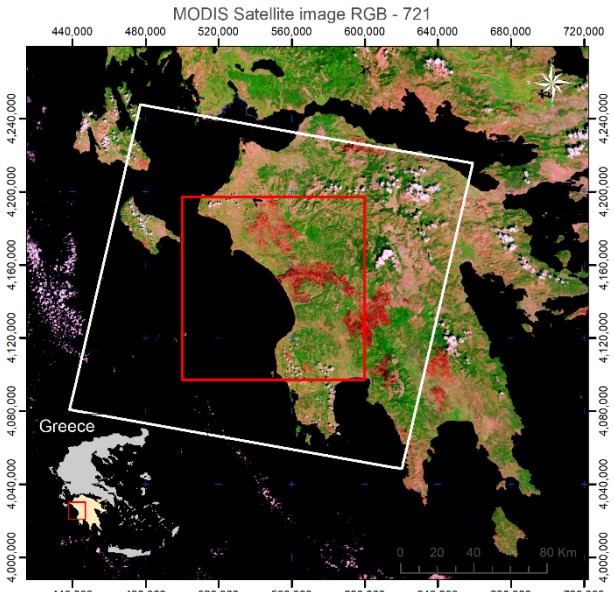

**Figure 2.** The study area was the Peloponnese, Greece (left), which experienced large wildland fires in 2007 that have been recognized as the most extreme natural disaster in the country's recent history [41], as seen in the MODIS image ("NASA/GSFC, MODIS Rapid Response") (right). The box in red is the square of 100 × 100 km centered in each Landsat scene used to reconstruct the recent fire history. The coordinate system is the Greek Grid (EGSA87).

This sampling design created sampling plots that are representative of all biomes considered, although, in this paper, we use the Mediterranean as a pilot study to show the whole methodology as captured by the study area of the Peloponnese, Greece.

### 2.1.2. Study Area

The Peloponnese (Figure 2), located in the southern part of Greece, covers an area of approximately 21,549 km$^2$, with 638,942 people living in the area, whereby most of them are employed in agricultural activities. Various vegetation zones associated with specific climatic zones are present, because of the diverse terrain, the altitude of which ranges from sea level to 2407 m. The western part of the Peloponnese is more wet than the eastern one since higher annual precipitation is usually observed in the western part. The main land-cover categories in the Peloponnese are dense and sparse forests (Sulla-Menashe and Friedl, 2019) according to the MODIS/Terra + Aqua Land Cover Type Yearly L3 Global 500 m product for 2018. A detailed description of vegetation categories, associated with the diverse topography featuring the typical Mediterranean Aleppo pine (*Pinus halepensis* Mill.) forests and the phryganic and evergreen sclerophyllous shrublands in the lowlands, can be found in Koutsias et al. (2012). Fire activity is evident in the Peloponnese, experiencing the biggest fires ever recorded in Greece in the year of 2007 [41].

### 2.1.3. Satellite Remote Sensing Data

The three research objectives we set up in this work were examined exclusively on the basis of satellite data at various spatial and temporal scales by using the spectral information content of the original spectral channels and estimated vegetation indices. The basic assumption is that there are changes due to fire in the vegetation and ground (destroy of vegetation, ash disposal, etc.), and these changes are reflected in the spectral information content and in the estimated vegetation indices. Phenology allows for the estimation of the time that the signal after the fire becomes similar to the signal before the fire and, thus, might be considered as an indication regarding the level and the time of the recovery.

The satellite data used in the current work are composed of Landsat, Sentinel-2, and MODIS satellite images. Their spectral characteristics are depicted in Figure 3, where it is obvious that there is a spectral consistency across the sensors, since their spectral bands correspond to the same or similar parts of the electromagnetic spectrum. Concerning their differences in spatial and temporal resolution, it seems that these do not create any incompatibilities in the research objectives of our study, since each satellite is used for a different research question. Firstly, Landsat is used to reconstruct the fire history from 1984 that guarantees an extended period being able to characterize the fire regime. Secondly, Sentinel-2 is only used to show that the rule-based approach can be used successfully to map the burned areas, thus guaranteeing that fire mapping can be implemented in future with Sentinel-2 images given the possibility of capturing smaller fires because of the higher spatial resolution. Lastly, MODIS images are used only to characterize vegetation phenology in the pre-fire situation and post-fire phenology patterns of fire-affected areas, taking advantage of the high-temporal-resolution data, which are important for the phenology.

### Spectral resolution of Landsat-8 OLI, Sentinel-2 & MODIS satellites

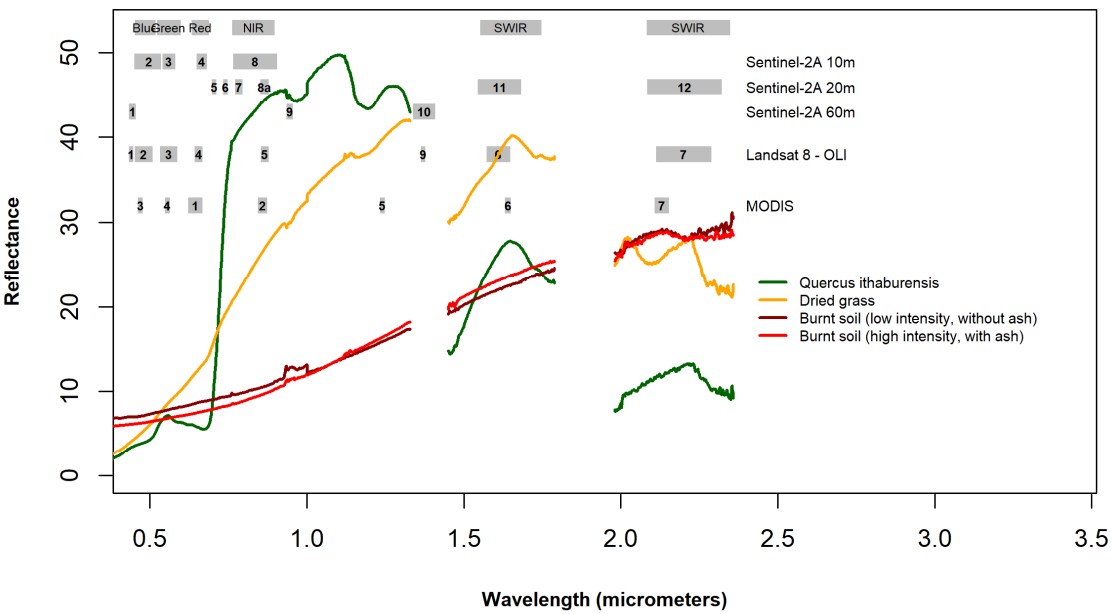

**Figure 3.** Spectral channels of Landsat, Sentinel-2, and MODIS sensors. The numbers in the grey boxes correspond to the numbers of spectral channels of each satellite.

**Landsat satellite images:** The main satellite data used in the current work are Landsat Level-2 archived data of surface reflectance for the period 1984–2015, provided free of charge by the US Geological Survey (USGS), and processed to Standard Terrain Correction (Level 1T). Landsat data are systematically corrected (i.e., radiometrically and geometrically) by incorporating ground control points while employing a digital elevation model (DEM)

(http://landsat.usgs.gov (accessed on 1 September 2017)). Therefore, we did not proceed to perform any further corrections to eliminate errors induced by different sources including the irregular terrain. These data were ordered and downloaded free of charge from the USGS for all Landsat series sensors (5, 7, and 8).

**Sentinel-2:** Sentinel-2 satellite images were also used mainly to check the methodology that we developed to map burned areas originally using Landsat images with the rule-based approach [15,42]. The Copernicus Sentinel-2 mission consists of two sun-synchronous satellites providing satellite data every 5 days. There is a certain improvement on some characteristics compared to Landsat, and it also provides data in very similar spectral regions to those of Landsat. This gives the opportunity for a very general spectral consistency between the two sensors, allowing the application of techniques, originally developed for Landsat data, to Sentinel-2 data without significant modifications.

**Surface Reflectance 8-Days MODIS:** According to MODIS (or Moderate Resolution Imaging Spectroradiometer) information, MODIS Surface Reflectance products can be an estimate of the surface spectral reflectance at ground level that is free of atmospheric scattering or absorption (https://modis.gsfc.nasa.gov/data/dataprod/mod09.php (accessed on 1 September 2017)). The 8-days products are composed by the best possible L2G observation during an 8 day period, as selected on the basis of high observation coverage, low view angle, the absence of clouds or cloud shadow, and aerosol loading. All available 8-days composites from the beginning of the operation of both satellites Terra (2000–) and Aqua (2002–) were downloaded from the USGS.

2.1.4. Vegetation Indices

A vegetation index (VI) is usually a linear transformation of two or more spectral bands aimed at enhancing the spectral signal of the original spectral channels by creating a new spectral space. This is sensitive to variations of vegetation attributes such as leaf area index (LAI), percentage green cover, chlorophyll content, green biomass, and absorbed photosynthetically active radiation (APAR). In our study, we chose a few that are considered typical and have been extensively applied for vegetation studies or in remote sensing of wildland fires. It is beyond the scope of this study to make an extensive evaluation of vegetation indices concerning their phenology between low and high fire-prone areas. The vegetation indices selected are described below.

**Normalized difference vegetation index (NDVI):** The normalized difference vegetation index (NDVI) [43,44] is one of the most used and well-known vegetation indices that is sensitive to live green plants in multispectral remote sensing data. Conceptually, the index is based on (a) the absorption of visible light, especially in the range 0.4 to 0.7 μm (red—MODIS band 1) by the pigment in plant leaves, and (b) the reflection of near-infrared light, in the range 0.7 to 1.1 μm (NIR—MODIS band 2) because of the cell structure of the leaves. The index, which varies between −1 and +1, minimizes topographic effects.

$$NDVI = \frac{NIR - Red}{NIR + Red} \tag{1}$$

where Red is MODIS band, 1 and NIR is MODIS band 2.

**Ratio vegetation index (RVI):** The ratio vegetation index (RVI) [45] simply divides the near-infrared (NIR—MODIS band 2) by the visible spectral values (red—MODIS band 1). Both RVI and NDVI measure the slope of the line between the origin of red–NIR space and the red–NIR value of the image pixel.

$$RVI = \frac{NIR}{Red} \tag{2}$$

where red is MODIS band 1, and NIR is MODIS band 2.

**Normalized burn ratio (NBR):** The normalized burn ratio (NBR) index [46,47], which is a modification of the normalized difference vegetation index (NDVI), was originally proposed by Lopez-Garcia and Caselles [48], who underlined the post-fire radiometric



changes occurring in the shortwave infrared part of the electromagnetic spectrum that was later verified by Koutsias and Karteris [49]. The NBR index results from NDVI by replacing red with SWIR, which is sensitive to leaf water content because of the absorption of electromagnetic energy at this wavelength [50]. There is a unique spectral behavior of burned areas in the NIR and SWIR region of the electromagnetic spectrum as described in Koutsias and Karteris [49,51]. In many burned land mapping studies, NBR, which attempts to maximize reflectance change due to fire [52], was found to be very effective.

$$NBR = \frac{NIR - SWIR}{NIR + SWIR} \tag{3}$$

where SWIR is MODIS band 7 and NIR is MODIS band 2.

**Normalized difference water index (NDWI):** The Normalized difference water index (NDVI) proposed by Gao [53] is a spectral vegetation index sensitive to the water content of plant leaves. NDWI is calculated using the spectral channels that correspond to near-infrared (NIR—MODIS band 2) and shortwave infrared (SWIR—MODIS band 6).

$$NDWI = \frac{NIR - SWIR}{NIR + SWIR} \tag{4}$$

where SWIR is MODIS band 6 and NIR is MODIS band 2.

**Shortwave infrared water stress index (SIWSI):** Similar to the normalized difference water index (NDWI), the shortwave infrared water stress index (SIWSI) make use of the spectral information of NIR and SWIR channels of electromagnetic spectrum being sensitive to water content [54]. Two configurations for this index have been proposed depending on which SWIR channel from MODIS data is used: SIWSI (5,2) and SIWSI (6,2) when MODIS channels 5 and 6 are used, respectively. Since band 6 is used in the calculation of NDWI, here we use only band 5.

$$SIWSI = \frac{NIR - SWIR}{NIR + SWIR} \tag{5}$$

where SWIR is MODIS 5, and NIR is MODIS band 2.

*2.2. Methodology*

2.2.1. Reconstruction of Recent Fire History and Determination of Fire Regimes

The spatially explicit reconstruction of the recent fire history and the determination of fire regimes consists firstly of the delineation of the fire perimeters (using time series of Landsat (1984–2015) satellite images), as well as of the unburned vegetation patches within it. To delineate the perimeters of the burned areas, the rule-based approach, developed by Koutsias et al. [15] and modified also by Koutsias and Pleniou [42], was applied to the available series of satellite images. This approach is a recently presented method, based on a semi-automatic algorithm for developing and applying rules on the basis of the spectral characteristics of burned areas in pre- and post-fire satellite images [15]. All images in the time series were processed by pairs of two images consisting of one pre-fire and one post-fire image, and new fires appeared in the post-fire images were properly delineated by the method. The final results were then cleaned manually by the user since commission errors were also found in the final results. The method, shown in Figure 4, although developed for Landsat, also works properly with Sentinel-2 without any modification due to the spectral consistency of the two sensors.

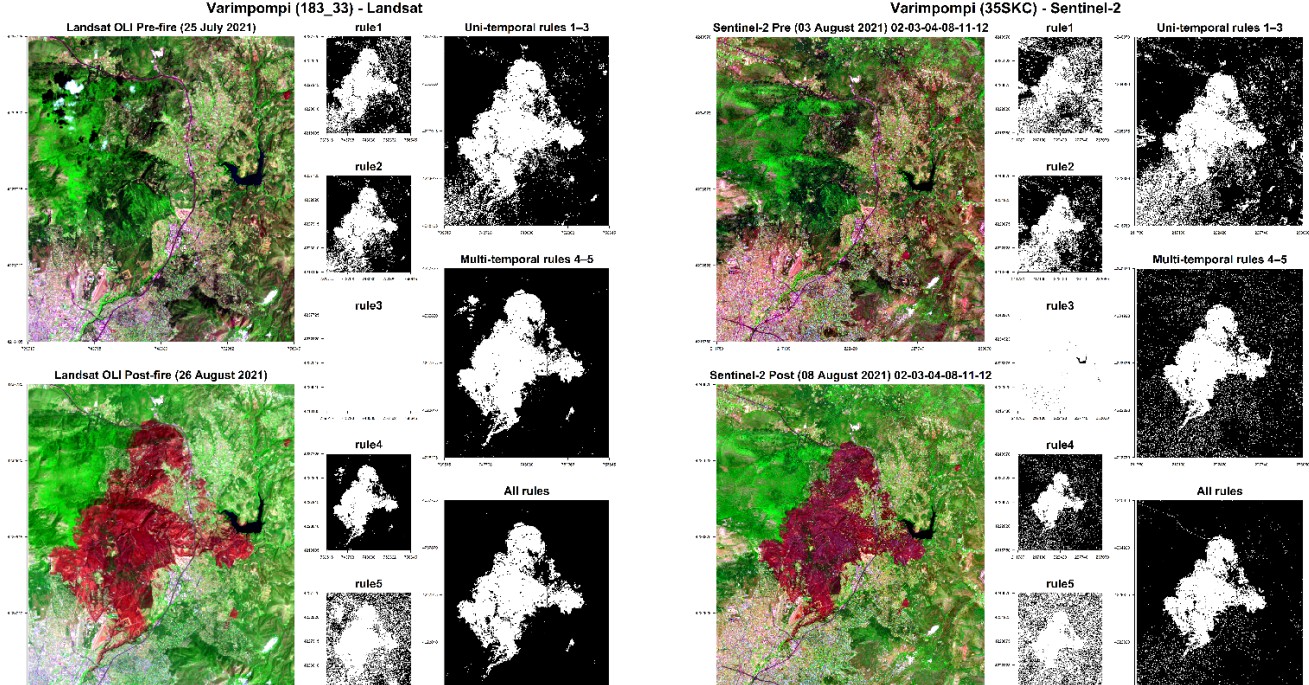

**Figure 4.** The modified rule-based approach as applied to map a very recent fire that occurred in 2021 in Attica, Greece using both Landsat pre- and post-fire satellite images (left) and Sentinel-2 pre- and post-fire satellite images (right). This is to show that the rule-based approach can also be applied to map the burned areas using Sentinel-2 satellite images.

After the reconstruction of the recent fire history, the next step was the application of the appropriate spatial analysis in a geographic information system environment to develop thematic layers that summarize the spatial and temporal information of fire regime characteristics, such as burned area maps, frequencies of fire events, and fire return interval maps.

### 2.2.2. Vegetation Phenology in the Pre-Fire Situation

For each fire incident, time series data of the reflected electromagnetic radiation as recorded by each spectral channel of the satellite sensor were created, and the above selected vegetation indices were calculated. We also calculated averaged values of each spectral channel and for each spectral vegetation index in an annual basis in order to estimate the parameters describing the annual vegetation phenology before the fire [14,55,56]. The analysis was restricted, however, only to fires that occurred after 2002 since MODIS data are available only after 2000. The average values and average $\pm$ 1 standard deviation values were estimated using all time-series observations from the beginning (2000) to the year just before the fire occurrence. The analysis of vegetation phenology was also restricted by fire size because the MODIS images had a resolution of 250–500 m for the satellite products we used; therefore, only fires exceeding a threshold could be considered in the analysis.

### 2.2.3. Post-Fire Phenology Patterns of Fire-Affected Areas

For the delineated fire perimeters, a phenological pattern was created by analyzing the time series of each spectral channel and each vegetation index. The phenology of the areas before and after each fire incident was compared by applying the dynamic time warping (DTW) method [57–59]. The DTW is a powerful method applied when two different time series are compared to identify similarity patterns. Such a comparison results in very poor conclusions regarding similarity when the simple and well-known Euclidean norm is used, thus underlying the necessity of a more complex and sufficient tool. A more elastic nonlinear alignment provides unconscious results for the similarity patterns, allowing similar shapes to match even if they are out of phase in the time axis. This may be obtained

through the minimization of the total "distance" between the points of the two datasets. In this case, DTW is a powerful method that compares the *i*-th point of the first dataset with the (*i* + 1)-th and (*i* − 1)-th points, respectively, while sometimes the (*i* + 2)-th and (*i* − 2)-th points are also considered. Such a procedure is able to identify similar shapes embedded in phase-delayed datasets. Such a function should be monotonic and continuous.

Given two datasets, $A = \{a_1, a_2, a_3, .., a_n\}$ and $B = \{b_1, b_2, b_3, .., b_m\}$, the DTW method is based on the calculation of the cost matrix M, which is defined as follows [60]:

$$M_{i,j} = |A_i - B_j| + min\{M_{i-1,j-1}, M_{i,j-1}, M_{i-1,j}\} \tag{6}$$

The above relation produces a quality indicator for the warping path we are interested in; a lower cost value indicates a closer-to-optimal DTW path. The cost matrix is then used to identify the warping path. The starts from the top right corner of the matrix and moves toward the bottom left, according to the simple rule of "moving to the left, down, or diagonal neighbor with the minimum cost value". This sequence produces a path through the cost matrix, whose quality is confirmed by the final normalized distance D given as the summation of the cost values of the specific cells corresponding to the path, divided by the number of these cells. As this value represents the discrepancy between the two datasets, it is rather obvious that a closer value to zero denotes a higher similarity between the two datasets.

In terms of an algorithm, the above procedure can be described as follows:

Step 1: Consider the two datasets *A*, *B*.

Step 2: Create the cost matrix *M*. Its rows and columns stand for the two sets to be compared. The cell values are calculated through the cost function expressed above.

Step 3: Identify the warping path by following the rule of minimum cost.

Step 4: Calculate the final normalized distance.

Lastly, it is worth noticing that the above algorithm can be applied to more than two datasets, where the matrix becomes a tensor, whose third dimension is equal to the number of compared sets. An example to illustrate the algorithm with phenology data from a burned area as compared to the pre-fire situation is provided in Figure 5. All computations were performed in R [61] using the "dtw" package [62].

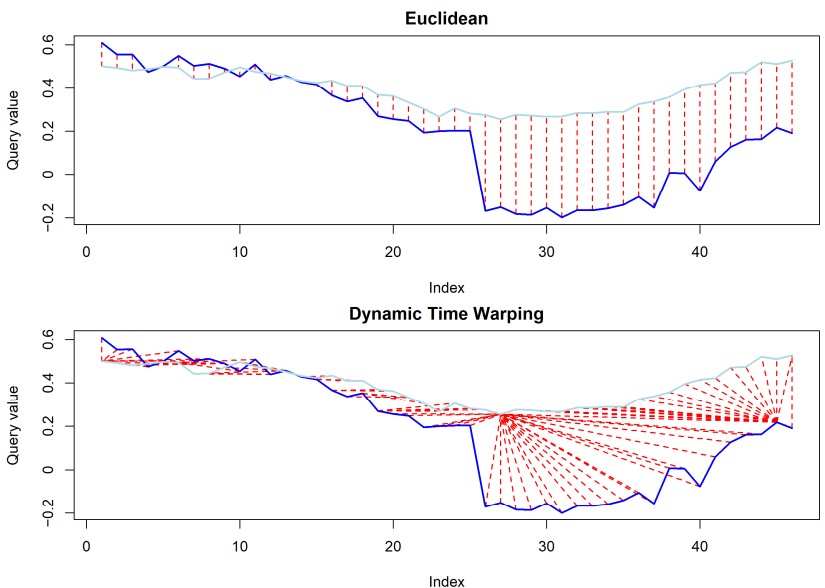

**Figure 5.** The phenology of the areas before and after the fire can be compared to each other by applying the dynamic time warping (DTW) method. All computations were performed in R [61] using the "dtw" package [62]. Dark-blue lines correspond to the reflectance of a burned area, while light-blue lines correspond to the reflectance of a vegetated area. Both lines were extracted from the 8-days composites within a year (46 observations in total) and are presented here as an example. Red lines are the one-to-one match for the Euclidean matching and the one-to-many match for the dynamic time warping matching.

## 3. Results

### 3.1. Reconstruction of Recent Fire History and Determining the Fire Regime

#### 3.1.1. Pilot Study Area—The Peloponnese

The first task of this research concerned (i) the spatially explicit mapping of burned areas for the period 1984–2015 (Figure 6) from Landsat and Sentinel-2 satellite observations using the rule-based approach [15] as modified and improved by Koutsias and Pleniou [42], and (ii) the determination of fire regimes (Figure 7). When this technique was applied to reconstruct the recent fire history in Attica, Greece, which is a similar area to the Peloponnese, within the period 1984–1991 and 1999–2009, a total of 1773 fire events were identified, and their scars were mapped. Most of the missing fire events corresponded to the 0–1 ha burned area class, followed by the 1–5 ha class, which is the next lowest. This is explained by the fact that small burned areas recorded by forest authorities may not always be captured by satellite data due to limitations of the spatial resolution of the sensor (i.e., 30 m for the case of Landsat) or due to limitations arising from the temporal resolution of the satellite [15].

**Figure 6.** Spatially explicit reconstruction of the recent fire history using Landsat images and the rule-based approach [15], as also modified and improved by Koutsias and Pleniou [42].

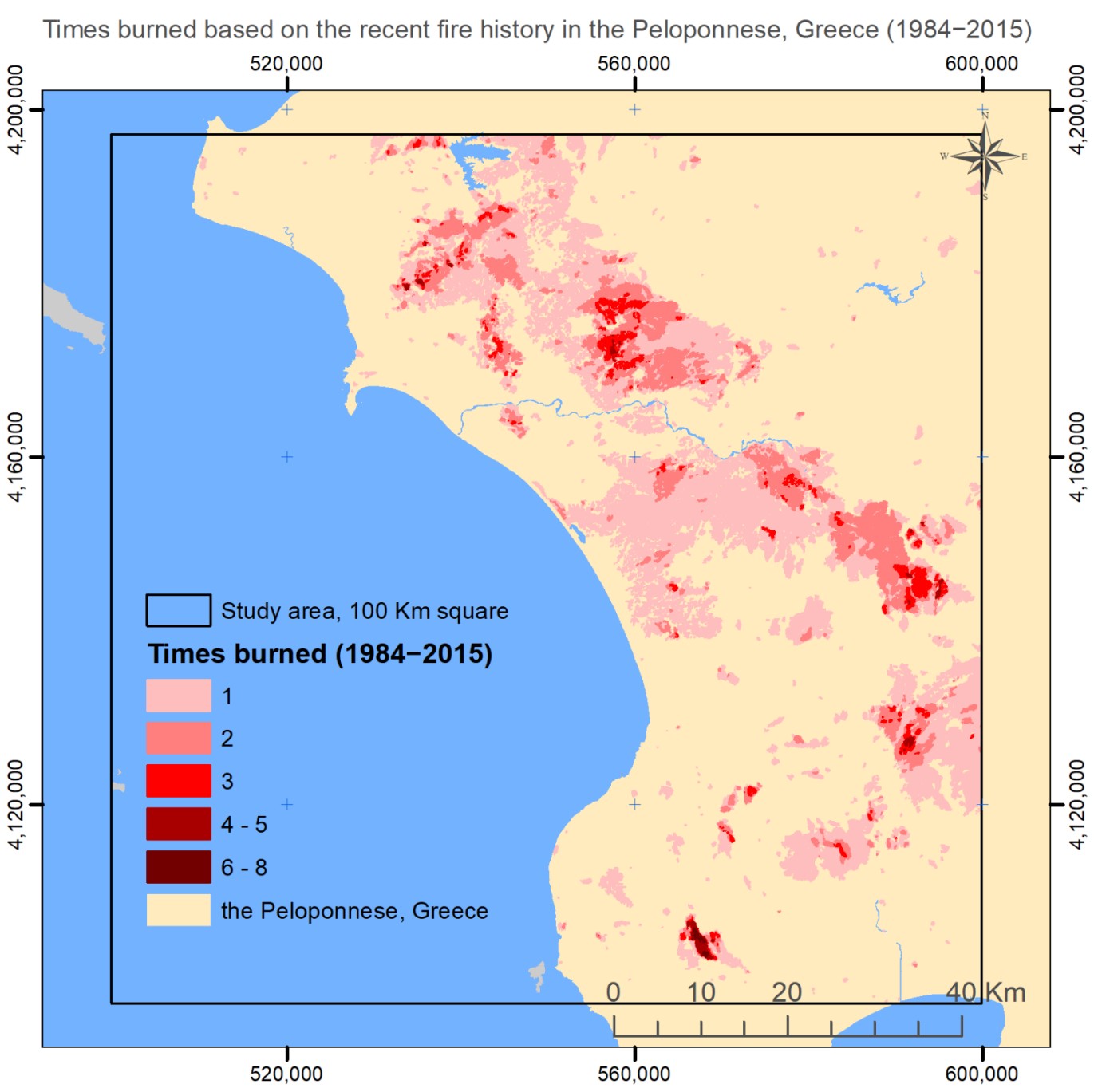

**Figure 7.** Times burned based on the spatially explicit reconstruction of the recent fire history using Landsat images.

3.1.2. Total Burned Area Per Biome

Total burned area statistics per biome were estimated after overlaying the 14 biomes (Figure 1) with MODIS fire perimeters (MCD64A1 Version 6.1 Burned Area). The MCD64A1 Version 6.1 Burned Area is a monthly 500 m pixel burned-area product available from 2000 to now. The summary statistics after the overlay of 14 biomes to the burned area product are presented in Figure 8, where it is obvious that the fires at global level occurred mostly in the tropical and subtropical grasslands, savannas, and shrublands (70%), followed by tropical and subtropical moist broadleaf forests (7%), and deserts and xeric shrublands (6.5%).

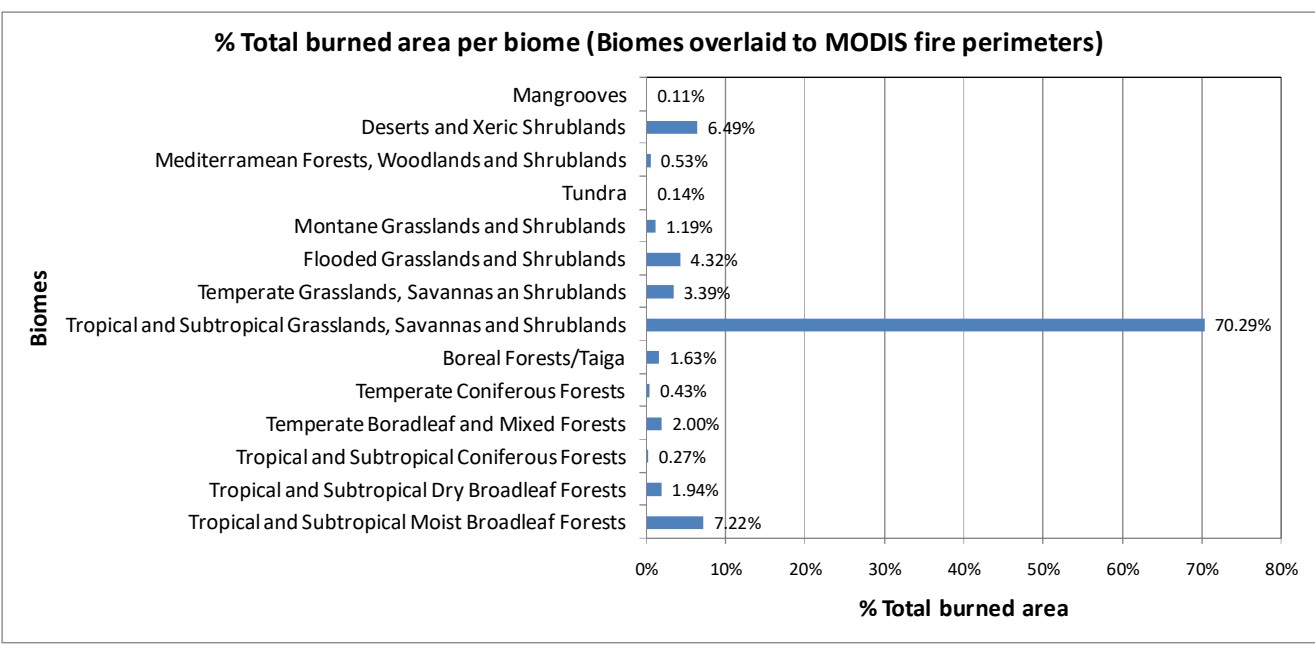

**Figure 8.** Total burned area per biome in percentages estimated after overlaying the biomes with MODIS fire perimeters.

The same summary statistics for each biome were also calculated also for the sampling areas corresponding to each biome (Figure 9). From the comparison of the summary statistics of both figures, it seems that there is a general consistency, while some minor differences can be observed. The most important differences can be observed for the fires at the biome montane grasslands and shrublands, with an increase from 1.19% to 12.93% in the sampling summary statistics, the fires at the biome Deserts and Xeric Shrublands (with a decrease from 6.5% to 1.6% in the sampling), and the fires at the biome tropical and subtropical moist broadleaf forests (with a decrease from 7.2% to 1.2% in the sampling).

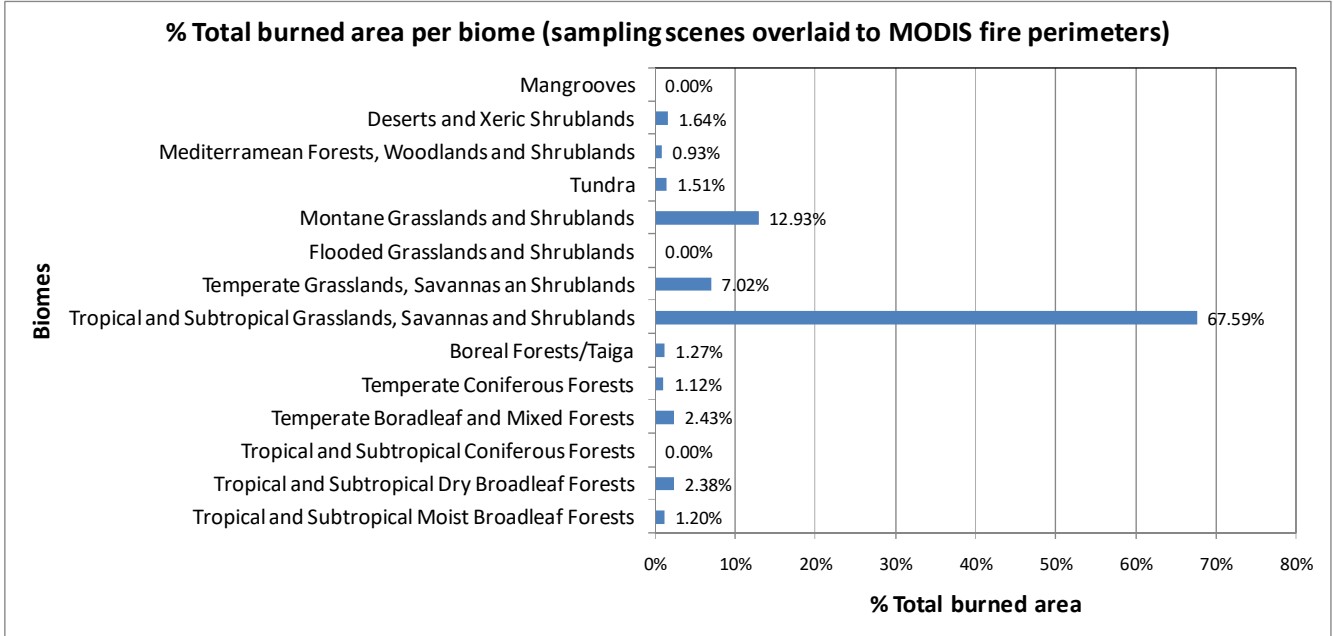

**Figure 9.** Total burned area per biome in percentages estimated after overlaying the sampling scenes with MODIS fire perimeters.

Similarly, the same summary statistics for each biome were also estimated for the sampling areas corresponding to each biome, but this time overlaid with Landsat fire perimeters as mapped using the rule-based approach (Figure 10). A general consistency can be observed, again with a few discrepancies. The most important were the fires at the biome montane grasslands and shrublands, with an increase from 12.93% in the sampling summary statistics of the MODIS fire perimeters to 25.25% in the sampling summary statistics of the Landsat fire perimeters. The fires at the biome tropical and subtropical moist broadleaf forests of the Landsat fire perimeters increased to 9.73% from 1.2% in the MODIS fire perimeters, being similar to the original 7.22%. It was beyond the scope of the study to conduct an extended evaluation and comparison of the summary statistics.

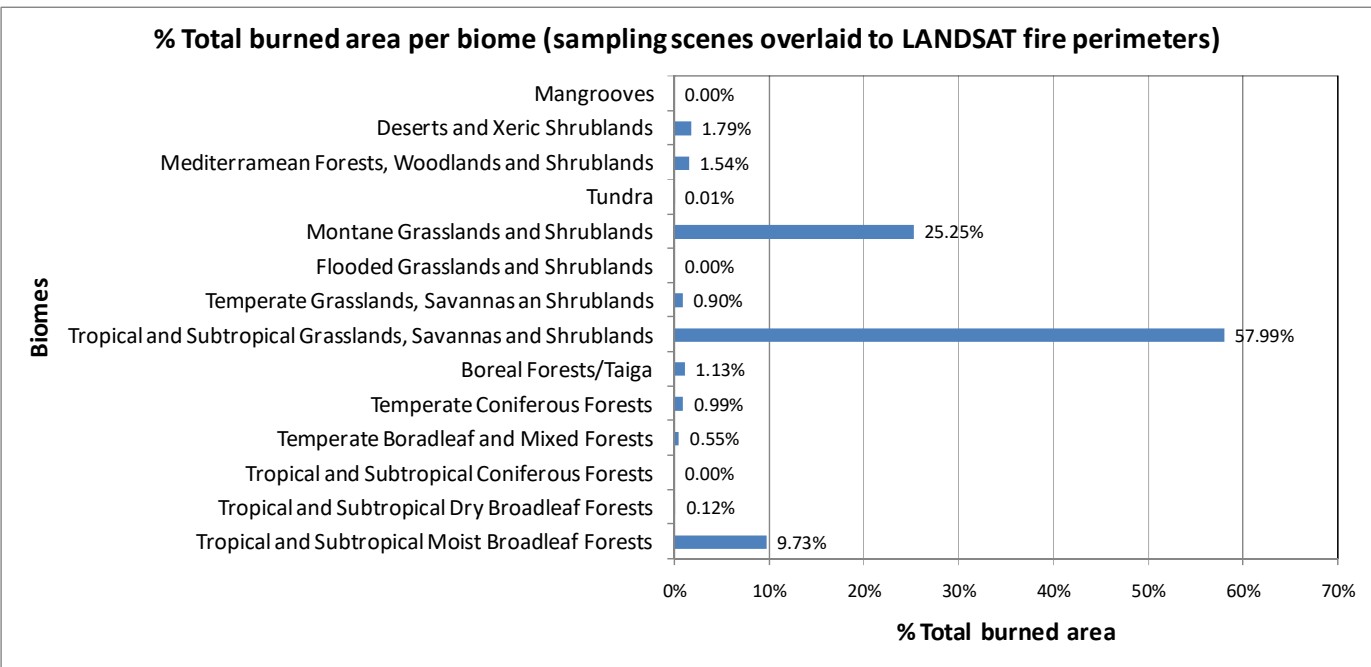

**Figure 10.** Total burned area per biome in percentages estimated after overlaying the sampling scenes with Landsat fire perimeters as mapped using the rule-based approach.

### 3.2. Vegetation Phenology of the Fire-Affected Areas

The second task of this research concerned the assessment of vegetation phenology in the pre-fire situation using satellite observations. The phenology profiles using the 8 day time intervals provided by the MODIS 8-days composites are presented in Figure 11, while the same profiles but aggregated to monthly intervals are presented in Figure 12. The phenology profiles show specific patterns of the vegetation before the fire in the fire-affected areas. These graphs show the spectral vegetation profile of the areas that are going to be burned. The average (red solid line) and average $\pm$ 1 standard deviation lines (dashed lines) were estimated using all time-series observations from the beginning (2000) to the year just before the fire occurrence.

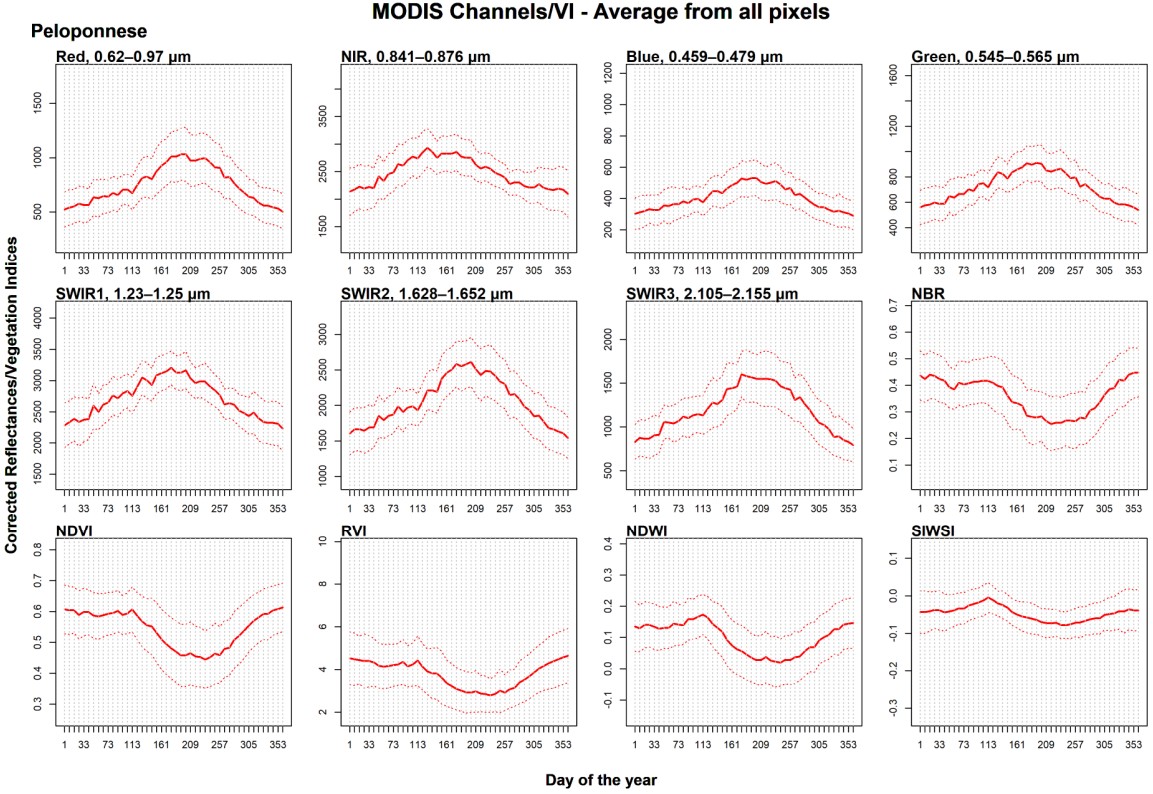

**Figure 11.** Phenology profiles using the 8 day time intervals provided by the MODIS 8-days composites, showing specific patterns of the vegetation before the fire in the fire-affected areas.

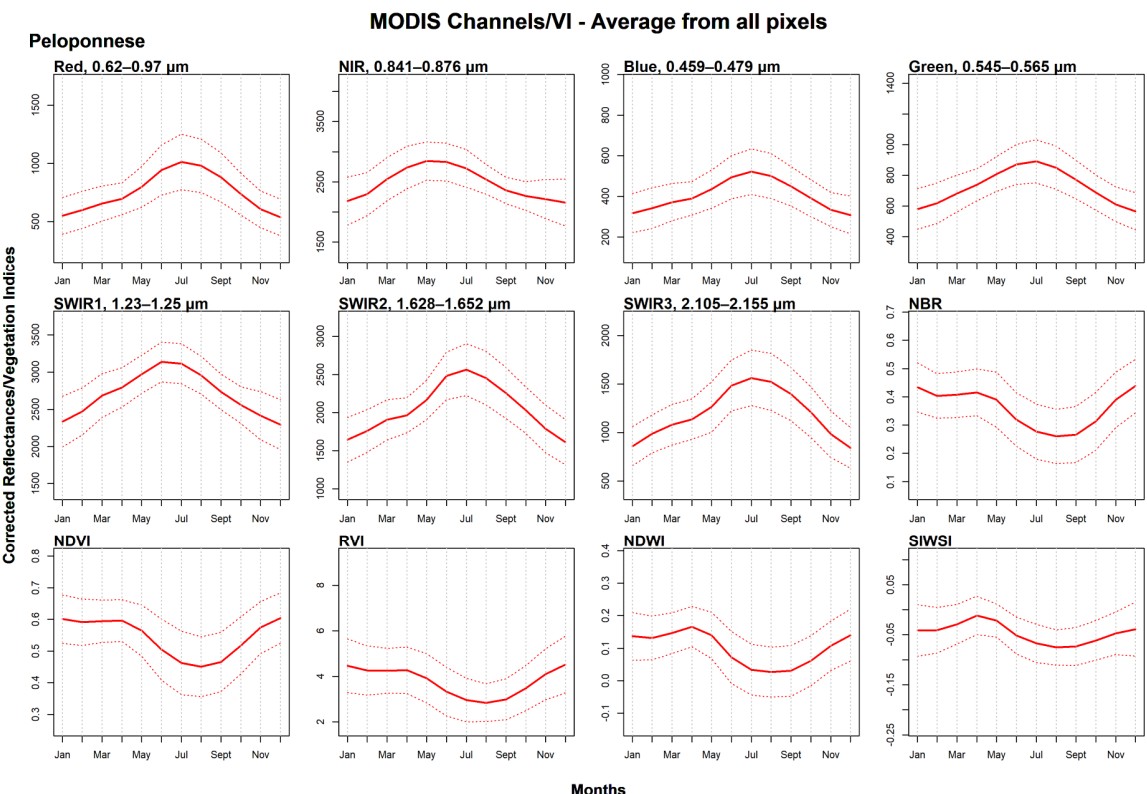

**Figure 12.** Phenology profiles using monthly time intervals, showing specific patterns of the vegetation before the fire in the fire-affected areas.

### 3.3. Monitoring of Post-Fire Evolution Patterns

The third task of this work concerned the assessment of the phenology of burned areas with MODIS satellite observations for the monitoring and comparative evaluation of the post-fire recovery patterns in the selected biomes and its comparison to the pre-fire situation using the dynamic time warping method (Figure 13). Here, we present only the vegetation indices considered for a single fire (fire ID 1942). To avoid any misunderstanding, the blue line was inserted by the yearly average phenology profile estimated by the observations from the beginning of the time series until the previous year of the fire. By comparing the DTW distance of the fire-affected area (red line) with the corresponding one of the pre-fire situation (blue line), we estimated the time period that the area needs to recover as this is defined by the spectral matching of the two phenology patterns.

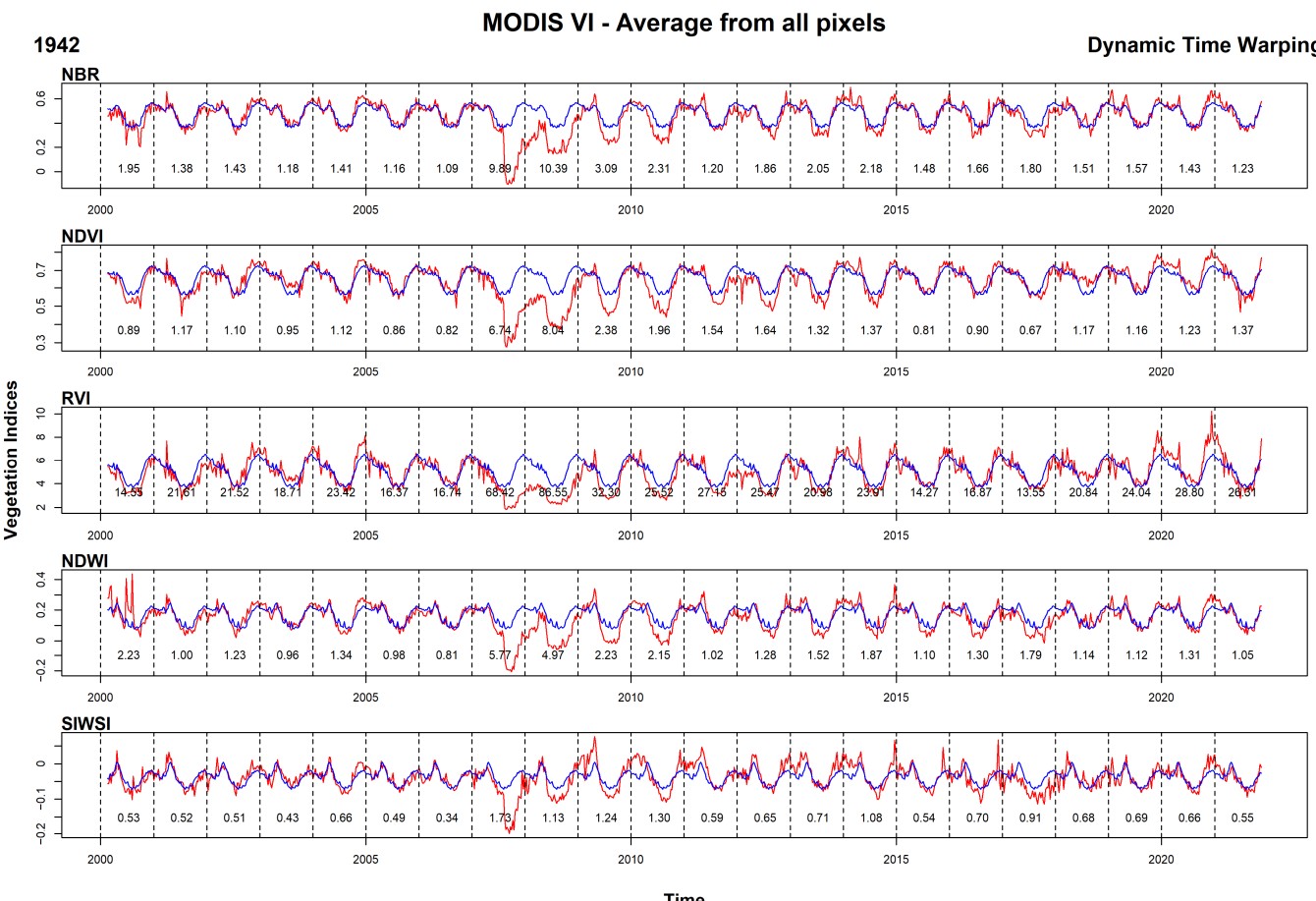

**Figure 13.** For the delineated fire perimeters, a phenological pattern of the burned areas was created, as a result of the analysis of the time series of the radiometric values of each spectral channel and each vegetation index. The phenology of the areas before (blue line) and after (red line) each fire incident was compared to each other by estimating their distances with the dynamic time warping method.

## 4. Discussion

This study showed that satellite images from Landsat and Sentinel-2 can be used successfully to reconstruct the recent fire history by applying an appropriate mapping technique. The rule-based approach applied here is composed of a set of spectral-based rules that are valid, especially when the post-fire image is captured shortly after the fire. This method tries to minimize the human involvement by enabling the algorithm to run automatically for a time series of satellite images that otherwise would need extensive time investment. The proposed method is based on the elimination of the time-consuming phase that is needed to train the algorithm. This is critical when many satellite images must

be processed, such as for the spatially explicit reconstruction of recent fire history, where thousands of images might be used in the processing chain [15].

Unburned patches, or areas of low fire intensity, can be observed within the fire scar perimeter, similar to Pleniou et al. [16], where a diverse mosaic was also observed. This is a very common pattern in most wildland fires [63], and it is important in ecosystem functioning that might directly or indirectly influence early vegetation [64] and faunal post-fire dynamics [65]. Therefore, detailed fire scar maps including patterns of burned and unburned patches provide important information needed to (i) study landscape wildfire dynamics, (ii) understand and predict the recovery processes of fire-affected areas, and (iii) implement post-fire ecosystem management plans, among others.

The map presented in Figure 7, similar to Pleniou et al. [16], shows a mosaic of various fire frequencies ranging from 1 to 8, as observed in other cases where this method was applied (e.g., in Attica). Fire frequency (defined as number of fires per unit time [66]) and fire return interval (defined as the time until a given location reburns [67]) are important variables with a significant role in vegetation diversity and landscape heterogeneity. The same authors also mentioned that, under specific conditions, short fire return intervals could have an important negative effect on regeneration and induce changes in the composition and relative abundance of species with specific life history [68,69]. Thus, maps of fire frequency can help to decide on proper restoration practices and vegetation enrichment of ecosystems subjected to high fire frequencies, as well as predict the long-term species composition of fire-prone ecosystems.

The fire regime of an area is mainly determined by the interaction of vegetation, climate, topography, local microenvironmental characteristics, especially when low spatial and time scales are considered, as well as by changes in the land cover/land use categories [1,70–72]. The fire regime description incorporates a number of fire characteristics, which are mainly related to the intensity, frequency, and size of fires [73,74]. There is a concern regarding the dominant factor shaping and determining the fire regime of an area [42], focusing on land use categories, climate, and human factors. A simple answer to the question stated above becomes increasingly difficult, as local fire regimes show great differentiation in the direction and extent of their changes [75]. Fire activity in an area can introduce spatial variability and, thus, change and affect landscape characteristics such as landscape fragmentation or connectivity.

Regarding the vegetation phenology in the pre-fire situation and the post-fire phenology patterns of fire-affected areas, it is evident that time series satellite images can be very useful and informative. Vegetation phenology is an important characteristic that can be used in vegetation monitoring, especially when time series of satellite images are available. Temporal profiles extracted from time series of MODIS images can be used to describe vegetation phenology that can be further used to monitor post-fire vegetation recovery in fire-affected areas. In our study, satellite images from Terra MODIS satellites were acquired for the period 2000–2021 and processed to extract the temporal spectral profiles for selected fire-affected areas. This dataset and time period, analyzed together with the time that these fires occurred gave the opportunity to create temporal profiles for some years before the fire.

As mentioned above, plant formations are characterized and differentiated on the basis of the dominance of functional plant groups [76]. Differentiation in the proportion of life forms gives each formation particular phenological patterns [77,78]. Phenology, i.e., the study of the periodicity of biological events, is a result of the long-term effects of climate, reacting to its changes [79]. The direct relationship of phenology with climatic parameters makes it an excellent tool for applications related to vegetation monitoring, especially when using satellite observations [13]. Phenology has been used to distinguish and model the spread of plant formations with comparable results to those produced by the application of bioclimatic models [80]. In our study, in addition to the original spectral data, we used commonly accepted vegetation indices, which are mainly found in vegetation studies, as well as in burned area mapping studies. In a recent study implemented by Pleniou and

Koutsias [81], it was observed that the original spectral channels, on the basis of which these indices are estimated, show a kind of sensitivity to external ecosystem parameters, e.g., to the spectral reflectance of the background soil. Additionally, the use of such indices is also justified by the sensitivity of spectral reflectance to different burn and vegetation ratios [81].

It is worth mentioning that the number of studies focusing on the long-term recovery of Mediterranean vegetation types after fire, as set by Kazanis and Arianoutsou [82], Baeza et al. [83], Lloret et al. [84], and Nioti et al. [17], is rather low, compared to studies dealing with short-term regeneration after fire. This is mainly due to the fact that these ecosystems usually return to their previous state after 10–12 years, as well as due to the difficulty in analyzing the process of long-term vegetation restoration after forest fires [85]. Plant biomes/climatic zones are associated, to a certain degree, with a specific type of fire regime, determined from the fire history of the area, and they result mainly from the synergy of the climatic conditions and the type of vegetation, together with human activities. Plant biomes also correspond to specific vegetation phenology types, a feature that can be used for various applications related to vegetation monitoring, especially when time series of satellite remote sensing images are available. The assessment of a fire regime that results from the spatially explicit reconstruction of fire history and the post-fire monitoring of the fire-affected areas can be studied using time series of satellite images with remote sensing methods.

## 5. Conclusions

The purpose of this work was to propose and evaluate a methodology to reconstruct the recent fire history and determine fire regimes, as well as to assess post-fire recovery patterns of burned areas using exclusively satellite observations by studying the phenology of the landscape with time series of satellite images. As a pilot study area, we used a fire-affected area in Greece, the Peloponnese, which belongs to the Mediterranean biome and, thus, sets the basis for the extension for the entire planet.

Satellite remote sensing technology, together with the availability of time series satellite images of high spatial and temporal resolution, provides an efficient way to reconstruct the recent fire history and monitor vegetation recovery. The rule-based approach seems to be an effective method for burned land mapping that minimizes the human involvement, by allowing the algorithm to run automatically for a time series of satellite images. The proposed method is based on the elimination of the time-consuming phase that is needed to train the algorithm.

Additionally, it was shown that vegetation phenology can be used to monitor vegetation recovery in the fire-affected areas, especially when time series of satellite images are available. The dynamic time warping method we applied provides a more elastic nonlinear alignment, allowing similar shapes to match even if they are out of phase in the time axis.

**Author Contributions:** Conceptualization, N.K.; methodology, N.K. and F.C.; writing—original draft preparation, N.K.; writing—review and editing, F.C., A.K. and F.N.; supervision, N.K.; project administration, N.K.; funding acquisition, N.K.; formal analysis, A.K., F.N., N.K. and F.C. All authors have read and agreed to the published version of the manuscript.

**Funding:** This research was co-financed by Greece and the European Union (European Social Fund—ESF) through the Operational Program "Human Resources Development, Education and Lifelong Learning 2014–2020" in the context of the project "Fire regimes and post-fire evolution of burned areas in selected plant biomes of the planet studying the phenology of the landscape with time series of satellite images" (MIS 5047152). The publication of this article has been financed by the Research Committee of the University of Patras.

**Data Availability Statement:** The Landsat and MODIS satellite images used in the study are available to download from the USGS (https://earthexplorer.usgs.gov/ (accessed on 1 September 2018 to 1 September 2019)). Sentinel-2 satellite images can be obtained from the ESA (https://scihub.copernicus.eu/dhus/#/home (accessed on 1 September 2018 to 1 September 2019)). Other relevant data can be obtained upon request from the authors.

**Conflicts of Interest:** The authors declare no conflict of interest.

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
