# Peer review of "Assessment of Fire Regimes and Post-Fire Evolution of Burned Areas with the Dynamic Time Warping Method on Time Series of Satellite Images—Setting the Methodological Framework in the Peloponnese, Greece"

_remotesensing, doi:10.3390/rs14205237_

Round 1
Reviewer 1 Report
The manuscript entitled “Assessment of fire regimes and post-fire evolution of burned areas with the dynamic time warping method on time series of satellite images – setting the methodological framework in the Peloponnese, Greece” constitutes an interesting study presented in a well written and structured manuscript. The manuscript reads well throughout and despite the multiple methods and datasets involved, the methodology is described in details so it can be repeated, which is one of the main intentions of the manuscript.
The introduction presents all the background information required for the reader to be able to follow the manuscript while the results are presented with a good combination of figures and text.
I don’t have any major concerns about this manuscript although I believe the discussion could be enhanced emphasizing more on the results obtained by the long term analysis of the vegetation phenology patterns and the significance of the study for global scale monitoring of fire effects on ecosystems. However, this is only a recommendation.
A few minor comments I have them highlighted in the manuscript and the authors may wish to consider when revising the manuscript.

Reviewer 2 Report
1.The abstract part is long. This is a mistake that many scholars often make when writing articles. It is recommended that the author shorten the abstract length to about 200 words.
2.This article uses modis, Sentinel 2 and landsat satellites as the main fire monitoring satellites, but the time and space scales of these three satellites are different. The author should mention how to unify the time and space scales in the article.
3.What do 1, 2, 3, etc. in Figure 7 represent? Does each number represent a time period? It needs to be clearly marked in the diagram.
4.The three satellite spatial resolutions used by this author are very rough, and the author should clarify the applicable fire category in the article (for example: very large forest fire, how much more than the burning area).
5.What does the abscissa in Figure 11 mean?
6.As far as I know, the modis satellite was launched in 1999 and put into use in 2000, but the author did not explain the fire monitoring and vegetation monitoring before 2000.
7.When the author introduced the spectral characteristics of vegetation, he did not clearly explain the meaning of the changes in the spectral characteristics, which would cause confusion to the readers.
8.The authors need to add to the conclusion and discussion the changes in vegetation over the years, echoing the changes in your spectral information, it is meaningless to detect only spectral information.
9.In the summary, the author talks about the drawing of the burning ground, but rarely mentions the change of spectral information. I don't understand what is the relationship between the two? Does it make sense for spectral information collection?
10.I look forward to the author's ability to describe in the article the changes in vegetation diversity in the selected study area, etc.
Reviewer 3 Report
My review of the article titled “ Assessment of fire regimes and post-fire evolution of burned 2 areas with the dynamic time warping method on time series of 3 satellite images – setting the methodological framework in the 4 Peloponnese, Greece”
General: The topic is very interesting and useful for monitoring fire ecology and post-fire vegetation management. The main problem is the way the article is written. Long sentences with more than two central messages are common obscure for easy understanding and reading flow. The use of inappropriate punctuation is frequent, again reducing the readability of the article. It needs a close revision and making sure these problems are fixed. The methodology is not clearly presented, the result lacks focus, and the conclusion is out of context.
Specific comments and suggestions
Line 65-67: biomes, ….. to organizing
Line 77-79: Avoid unnecessary use of commas which is prevalent throughout the article. Example (in reds): It is thus understood that the climatic parameters which determine the fire regime also determine the special characteristics of the various plant formations and, accordingly, the plant formations, in turn are characterized by different fire regimes.
Line 88-91: inappropriate use of commas is disturbing the flow of the messages. For example here “So far, relatively few studies [31-34], have investigated 88 fire regimes comparing different vegetation…….” you are saying the sentence could also be read without a phrase in reading. Does the sentence give a meaningful sense without the phrase in red?
Line 93-98: The following paragraph is a stand-alone paragraph which does not have any linkage to its previous or next paragraphs. It is not even clear why it is there.
“The availability, at no cost to the final user, (i) of Landsat satellite imagery by the US Geological Survey (USGS), (ii) of Sentinel-2 satellite imagery, provided by ESA and (iii) of MODIS (or Moderate Resolution Imaging Spectroradiometer) imagery provided by NASA/USGS, allows for a low-cost obtaining and processing of time series of satellite images (e.g. 1984-today), that under other circumstances would be a considerable cost that could be prohibitive to research.”
Line 105-106: What does “……..and definition of the fire regimes with satellite data” mean? What do you mean by definition in this context?
Line 112: instead of mentioning examples such as “……. (e.g. MODIS satellite images), mention what you actually used. Do this all over the article.
The same in lines 135-136 “(e.g. kernel density interpolation method [38])”, if this is an example, then what did you actually use/implement?
Line 146: ….. were reconstructed.
Line 148. the opened bracket needs to be closed somewhere.
Line 177-178: “Fire activity is evident in the Peloponnese while it has experienced in the year of 2007 the biggest fires recorded ever at Greece by that time [42].”
Line 178-190: Rewrite the following sentence for better clarity of the message. “their spectral characteristics are depicted in Figure 3, where it is obvious the spectral consistency between the sensors since their bands present a spectral.”
Methods: in this section, it would be excellent to focus on what you did. Explanation about products should be addressed either in the introduction or just indicate where these could be found. Be to the point and clearly present your procedure chronologically. Avoid repetition, such as available without cost, which is mentioned multiple times.
Line 325-326: “ DTW is a powerful method applied when data obtained by two different time series should be compared to identify similarity patterns.“Are the time series not the data by themselves? If not, what data are you obtaining from the time series? Make your messages clear to your readers.
Figure 5: you explained what the blue lines are but not the ones in Red.
Overall methodology: We are introduced that you used a rule-based analysis to delineate and analyse burned areas. But, it lacks clarity, and not much is said about the rules and their criteria in a way that helps others to replicate similar studies.
Results:
Line 373-385: The whole paragraph is explaining about the methods used, their strong sides and specific recommendation on how they should be used. This should have been well explained in the methodology section.
Line 386-387: “ a similar area to Peloponnese.” A similar area where? Which area exactly? If I am correct, your study area is “ Peloponnese”, and in your result section “, pilot study area” should be referring to your study area. However, the paragraph is about another area which is similar to “ Peloponnese”. Focus as an issue throughout the paper.
Line 387: “ a total amount of 1773 fires had been identified and mapped.” 1773 fires? What does this mean? 1773 times (frequency) or in 1773 places or pixels (spatial extent)?
Line 403: “Total burned area per biome”: Because this was not presented clearly in the methods, how was this result found? Is it by applying the rule-based DTW developed for the pilot study area implemented on the 14 biomes?
Line 404-405: “ Total burned area statistics per biome have been estimated after overlaying the 14 biomes (Figure 1) to MODIS fire perimeters (MCD64A1 Version 6.1 Burned Area).” How is this related to your methodology? What is the importance of having this included if it does not result from the method you addressed?
Line 308: What type of disturbance? How was that measured? if it is from other literature, we need citations and need to move to the discussion section.
Line 406: replace concerns by was or is depending on the tense type you used to write the article.
Line 446-456: This section contains messages that should have been included either in the introduction or methodology. In the results section, focus on your results. No need to explain how it was done and what it is. This should have been addressed in the introduction or methodology sections. Surprisingly, nothing is said about the results you found about the phenological changes before/after the fire.
Line 470-481: The same as the above comment. Figures are important to visualize processes, but they are not enough by themselves. You need to explain your results. You put graphs, but you did not try to show your readers.
Discussion:
The results are not well presented, or even it is not clear what the results are in accordance to the title/objective set. Therefore, there is a point to commenting on the discussion part.
Conclusion:
The conclusion is simply a summary of the introduction. In conclusion, you do need to summarize the results. Here, try to convey the overall impression of your article and its contribution to the existing literature and mountain forest monitoring.
In your conclusion, I read, “The purpose of this work is to propose and evaluate the methodology to determine fire regimes as well as patterns of post-fire evolution of burned areas using satellite observations exclusively by studying the phenology of the landscape with time series of satellite images.” If this is the objective, do you think you have addressed/answered it in your results or discussion?
Look at your title, “Assessment of fire regimes and post-fire evolution of burned areas with the dynamic time warping method on time series of satellite images – setting the methodological framework in the Peloponnese, Greece”
According to your title, your intention is to set a methodological framework for detecting fire occurrence and evaluate vegetation phrenology after the fire. Now, what is the framework you set? What did you really do to set the framework?

Round 2
Reviewer 3 Report
Dear authors,
congratulations for the extensive revision within this short period of time. The readability and placement of contents is much improved.
Still, spelling and long sentences need to be checked up. You can still improve your conclusion.
Overall, my impression on the paper is much better than the original manuscript.
Well done!
